# Investigating the Applicability of Ichthyoplanktonic Indices in Better Understanding the Dynamics of the Northern Stock of the Population of Atlantic Hake *Merluccius merluccius* (L.)

**Paula Alvarez [1],\*** , **Dorleta Garcia [2]** and **Unai Cotano [1]**

1    AZTI, Marine Research, Basque Research and Technology Alliance (BRTA), Herrera Kaia, Portualdea z/g, 20110 Pasaia, Spain
2    AZTI, Marine Research, Basque Research and Technology Alliance (BRTA), Txatxarramendi Ugartea z/g, 48395 Sukarrieta, Spain
\*    Correspondence: palvarez@azti.es

**Abstract:** Abundance indices are essential data for the application of stock assessment models to obtain fish abundance estimates. Abundance indices have usually been derived from fishery-dependent data, yet the increase in fisheries-independent surveys is now offering new opportunities for these calculations. In this study, we explored the usefulness of ichthyoplankton indices derived from scientific surveys in estimating spawning biomass. In addition, we also investigated whether the strength of the year–class of the commercial cohort of Atlantic hake, as a determinant, could be defined at an early life stage. We used samples collected during the triennial mackerel and horse mackerel egg surveys (MEGS), which cover the hake spawning area in the Bay of Biscay. The biomass indices were determined as the abundance of eggs in the early development stage (stage 1) when transformed into egg production (EP) from 1995 to 2019 in the months of March and April—which is considered a period of high spawning activity for hake in this area. Additionally, we built a metric for larval abundance and converted larval length into age. This was in addition to constructing a pre-recruit year-class index (YCI) while using the EVHOE bottom trawl abundance database for hake for the period of 1997 to 2016. The results of regression analysis of egg production and spawning stock biomass indicate that both parameters are significantly correlated ($r = 0.76$). By connecting the abundance of eggs and larvae in the adjoining stages, we are able to identify two periods of high mortality associated with the transition from "yolk-sac-first" to "feeding larvae" and "late larvae-YCI10", but we were unable to discover when the strength of the recruitment year–class is determined. As such, it appears that for the northern stock of hake, recruitment is established in the late juvenile stages.

**Keywords:** early life stages; ichthyoplankton indices; recruitment; hake biomass

## 1. Introduction

Biological monitoring for the purposes of identifying changes in fishery resources comprises two datasets: Fishery-dependent and fishery-independent [1,2]. The fishery-dependent data are characterized by long time series, wide spatial coverage all year round, and information on a large variety of target species. However, limitations do arise due to constraints imposed by management, such as spatiotemporal restrictions, selectivity, and the dynamics of the fleets that tend to concentrate in areas with the highest abundance. In contrast, fishery-independent monitoring is mainly reliant on expensive research programs conducted at sea over relatively short periods of time. Such survey data are of higher quality due to the fact that the processes of sampling and collection are scientifically designed and standardized [3]. A combination of both data sources will most likely provide better knowledge of the population of interest than the use of each dataset separately.

Atlantic European hake (*Merluccius merluccius*) is a demersal and bentho-pelagic species widely distributed in the Northeast Atlantic Shelf; furthermore, it is a major commercial demersal species in the Bay of Biscay [4]. The International Council for the exploitation of the Sea (ICES) recognizes the existence of two stocks: The so-called northern (ICES Division 3a, Subareas 2,4,6,7 and divisions 8a,b,d) and southern (ICES divisions 8c, 9a) stocks [5]. The northern stock is assessed using a length-based age-structured statistical model (SS3) established by the ICES Working group for the Bay of Biscay and the Iberian waters ecoregion (WGBIE).

Commercial catch and biological data (individual weight, maturity, and growth) are usually insufficient to allow estimating the abundance using stock assessment models. Moreover, the time series of the abundance indices are necessary to calibrate the model and obtain more accurate estimates [6]. In the stock assessment of the northern stock of hake, abundance data obtained from three contemporary scientific surveys are included: EVHOE (EValuation Halieutique Ouest de l'Europe), a French scientific survey that covers part of the Bay of Biscay and Celtic Sea and catches mainly hake individuals smaller than 25 cm; a Spanish groundfish survey that covers the Porcupine Bank and catches of principally hake individuals in the 40–75 cm range; and an Irish groundfish survey that covers part of the Celtic sea with catches mainly of individuals smaller than 30 cm. Thus, individuals longer than 75 cm are not covered by any abundance index, and this is one of the main problems encountered in fitting the hake stock assessment model [5]. An egg production index that is representative of the spawning stock biomass could be used to calibrate this fraction of the population and thus provide more accurate estimates.

For many pelagic spawners, egg and larval production can be related to spawning biomass [7,8]. Furthermore, indices of the annual spawning biomass are successfully used for the stock assessment of many species in Northeast Atlantic (NEA) waters, such as anchovy, sardine, mackerel, and horse mackerel. The application of this method has also been evaluated for hake [9], with the conclusion that the daily egg production method is potentially applicable to this stock. Furthermore, larval-stage dynamics may be used as early indicators of future recruitment potential in adult fish stocks and serve as leading indicators of spawning success or failure [10].

In NEA waters, long-term ichthyoplankton surveys exist for some species of pelagic fish. Two of the largest, the mackerel and horse mackerel egg surveys (MEGS) and the international herring larvae surveys (IHLS), started in 1977 and 1967, respectively. The former, which is carried out every three years, encompasses such a large spatial and temporal area that it covers the spawning period of numerous fish populations [11].

The strength of year–class in stock as a determinant of recruitment is the result of the survivors of the eggs spawned during its reproductive season. The critical period hypothesis proposes that the strength of year–classes, as a determinant of fate, is established in the early larval stage, shortly after yolk absorption [12]. During these early life stages, eggs and larvae exhibit high mortality rates [12–17], after which this rate declines. Although it has traditionally been accepted that recruitment is determined during the earliest life stages, this statement is not valid for most fishes. In plaice, for example, levels of recruitment may be poorly fixed at the egg or earliest larval stages [18–20]. In other taxa, such as pelagic clupeoids, recruitment levels in most years may be set during the late-larval stage or in the long pre-recruit, juvenile stage, e.g., [21–24]. Knowing when year–class strength is established represents a great challenge for fishery science, as it could be a useful index for application in estimates with respect to commercial stocks.

Hake is a batch spawner species with an extended spawning season [25]. Although the spawning activity of hake takes place throughout the year [26], in regard to the Bay of Biscay area, major spawning occurs from February to April [27,28]. In the west of the North Sea, spawning occurs from August to September [29].

Based on the time series of hake egg and larvae abundance from MEGS, the purpose of this paper is to propose an ichthyoplankton index based on hake egg density as a potential index for SSB for use in the calibration of the hake assessment model as well as to also

explore whether hake year–class strength can be defined at any life history stage using simple approaches, such as egg production and larvae dynamics.

## 2. Material and Methods

### 2.1. Collection of Samples

Most of the ichthyoplankton samples analyzed in this study are from the ICES triennial mackerel and mackerel egg surveys (MEGS), which cover the Northeast Atlantic waters from Portugal to Iceland between February to July. In the Bay of Biscay, the surveys are conducted in the spring, from March to May (Table 1). In 1995, two specific hake egg and larval cruises were also conducted in February and March in the Bay of Biscay using a central systematic sampling scheme different from the MEGS (Table 1).

**Table 1.** The main characteristics of the surveys analyzed in this study. In all cases, the studied area corresponds to the Bay of Biscay. B-60 and B-40: Bongo plankton nets with 60 and 40 cm of mouth diameter, respectively. G-III-VII: Gulf plankton net, model III or VII.

| Year | Vessel | Survey in Bay of Biscay | Period | Gear | No. Samples |
|---|---|---|---|---|---|
| 1995 | Investigador | 22 March–1 April | March | B-60 | 62 |
| 1998 | W. Herwig<br>C. Saavedra<br>Tridens | 15–31 March<br>21–22 April<br>21–30 April | March<br>April<br>April | Nackthai<br>B-40<br>G-III | 27<br>9<br>59 |
| 2001 | W. Herwig<br>Investigador | 31 March–11 April<br>11–18 April | April<br>April | Nackthai<br>B-40 | 77<br>44 |
| 2004 | Investigador<br>Endeavor | 24 March–11 April<br>27 April–7 May | March–April<br>April–May | B-40<br>G-VII | 75<br>44 |
| 2007 | Itsaslagunak | 2–22 April | April | B-40 | 44 |
| 2010 | Investigador | 23 March–14 April | March–April | B-40 | 37 |
| 2013 | A. Alvariño | 22 March–6 April | March–April | B-40 | 64 |
| 2016 | R. Margalef | 19 March–7 April | March–April | B-40 | 50 |
| 2019 | R. Margalef | 19 March–6 April | March–April | B-40 | 76 |

The basic sampling unit used in these surveys was a 0.5° by 0.5° rectangle with the sample taken at the midpoint of the rectangle [30]. Two types of sampling gears were deployed. Most samples were collected using either a modified Gulf-type high-speed sampler [31] fitted with a 250 μm mesh net or a 40 cm Bongo net with 335 or 250 μm mesh size. Samples in 1995, however, were collected in the center of 10 nm × 30 nm rectangles along transects perpendicular to the 200 m depth contours. At each station, a plankton haul was performed using a 60 cm Bongo net furnished with nets of 333 and 505 μm mesh size.

Double oblique tows were conducted from the surface to within 5 m of the bottom, or a maximum depth of 200 m, at a payout and retrieval speed of 20 m/min. The vessel speed was set at 2.5 knots when towing a Bongo or 4 knots when using a Gulf. Flowmeters were used during plankton tows for estimating the volume of filtered water. The plankton material collected in the 250 μm net was processed and preserved in a 4% buffered formaldehyde solution. More information regarding these procedures can be found in ICES [30].

In most years of this study, the sampled area extended from 46 to 48° N, whereas in the years 1995, 2001, and 2004, the southern area of the Bay of Biscay was also encompassed (Figures S1 and S2). Despite this, the differences in coverage did not seem to have significant effects on the abundance of hake eggs and larvae, due to the fact that the main spawning grounds for hake are usually located at northern 46° N (see Figures S1 and S2). However, to avoid any eventual effect due to the spatial variability of surveys, we selected—as the standard area—the area for the year with the lowest sampling coverage, which was 2010; this area extended from 46°15′ N to 47°45′ N.

### 2.2. Eggs Identification and Standardization of Samples

A SAT test was used to identify hake eggs. This test is based on the hydrophobic characteristic of its chorion [32]. This feature is commonly applied when identifying fish eggs in order to separate eggs of hake from other species. After that, the hake eggs were classified into four development stages according to Coombs and Mitchell [33]. Only eggs in stage 1 (the earliest identified stage) were used in the analysis.

Hake egg production was estimated using the procedure described in [30]. The number of fish eggs per sample was standardized to the number per m$^2$, using the formula described by Smith and Richardson [34]:

$$\frac{Eggs}{m^2} = \frac{\text{No eggs} \times \text{Sampler depth (m)}}{\text{Volume filtered (m}^3)} \tag{1}$$

The egg production per m$^2$ per day was calculated for each sampled station. This production was based on the observed number of stage 1 eggs, i.e., the time it takes an egg to pass through this stage at the temperature observed on the station.

The number of eggs per m$^2$ was converted to numbers per m$^2$ per day using the following formula [30]:

$$EP = \frac{\frac{Eggs}{m^2}}{day} = \frac{24 \times \frac{\text{egg}}{m^2}}{a \times T^{-b}} \tag{2}$$

where $T$ is the temperature (°C). The time the eggs spend in stage 1 was calculated from the formula [33]

$$Z = 1264 \times T^{-1.411} \tag{3}$$

where $Z$ is the time (in hours). $Eggs/m^2/day$ was then equalized to the area of the rectangle it represents using the following formula:

$$EP \times Area \left(m^2\right) \tag{4}$$

Rectangle areas were determined by each half-degree row of latitude using the following formula:

$$Area \ (m^2) = (\cos(\text{latitude}) \times 30 \times 1853.2) \times (30 \times 1853.2) \tag{5}$$

Total daily egg production for a cruise/period was calculated as the sum of individual daily egg production per rectangle.

The total and positive area (i.e., the area with a presence of stage 1 eggs) was calculated as the sum of each rectangle area for each period considered.

### 2.3. Larvae Identification and Standardization of Samples

Hake larvae preserved in 4% formaldehyde were identified and measured to the nearest lower 0.1 mm standard length (SL) using an ocular micrometer. Although formaldehyde reduces the length of the larvae by approximately 4.3% [35], no corrections to allow for larva shrinkage due to fixation procedures were made in this study. The shrinkage is assumed to remain constant if the preservation procedure remains unchanged and should not affect the study results.

The number of fish larvae per sample was standardized using Equation (1).

Hake larva length was converted to age by applying the length-at-age relationship established by Alvarez and Cotano [36]. We assumed the absence of inter-annual differences in the growth rate according to the variability in growth reported by other authors (Table 2, [36]) for the range of temperature observed during this study (11.3–12.6, see Table 3). The larvae were then gathered into day classes, and the abundance (number/m$^2$, Equation (1)) per station was calculated. Each larval abundance by age and station is

equated to the area it represents by multiplying this value by the square meter of each rectangle (Equation (4)) for obtaining the number of larvae for each age class.

**Table 2.** Larvae abundance index used in the analysis.

| Index | Age Class (Days) | | SL Class (mm) | |
|---|---|---|---|---|
| | Min | Max | Min | Max |
| N < 10 | 0 | 10 | 2 | 3 |
| N > 10 | 11 | 52 | 3.1 | 13 |
| N > 15 | 15 | 52 | 4.1 | 13 |
| N > 25 | 25 | 52 | 6.1 | 13 |
| Ntot | 0 | 52 | 2 | 13 |

**Table 3.** Year–class index used for the analysis.

| Index | Comment |
|---|---|
| Recruit_BIE | SS3-derived age-0 recruitment for NSH [37] |
| EVHOE | Juveniles index based on EVHOE bottom trawl surveys |
| YCI 10 | Abundance of hake < 10 cm based on EVHOE surveys |
| YCI 20 | Abundance of hake 11–20 cm based on EVHOE surveys |
| N (age) | Larval index for individual by age (see Table 2) |

Finally, for each year, the abundance of larvae was computed as the sum of larvae by age class according to the age ranges shown in Table 2.

$$Ncx = \sum_{age=0}^{age=xi} Nl(1) + Nl(2) + \cdots + Nl(xi)$$

Here, *Ncx* is the number of larvae belonging to each of the age classes "*cx*", as defined in Table 2 (0–10, >10, ... ). Furthermore, "*xi*" is age (in days).

### 2.4. Larvae Survival Estimates

Apparent larval mortality can be determined based on the decline in larval abundance by age or by length [38]. In this study, we opted to use the parameter length instead of age due to the fact that length is a less error-prone measure than age, and also because the models with length, as they are variable independent, produce more statistically significant results with similar temporal trends. For each year, the hake larvae were grouped into 1 mm SL classes. The data on hake larva abundance (number/m$^2$) at a standard length (mm) was fitted to an exponential decay model that is empirically expressed as

$$ABD = \alpha \times SL^{-Z(\pm SE)}$$

where *ABD* is the abundance of hake larvae (number/m$^2$), *SL* is the standard length (mm), $\alpha$ is the intercept, and $Z$ is the instantaneous mortality rate.

Mortality as a percentage was determined as

$$M = [1 - \exp(Z)] \times 100$$

where survival (SvR) is obtained as $1 - M$.

### 2.5. Year–Class Abundant Standardization Data

The assessment of the northern hake stock was conducted using the Stock Synthesis 3 (SS3) model [39]. The assessment includes two abundance indices for small individuals: EVHOE, a French survey conducted in the Bay of Biscay and Celtic Sea, and an Irish groundfish survey (IR_IGFS), which is conducted in the Irish Sea. We selected the

information collected by the former due to the fact that its coverage suitably fits the area analyzed in this study. Furthermore, this same survey has been conducted since 1997.

Using the EVHOE survey data stored in the ICES DAtabase of TRAwl Surveys (DA-TRAS), we built an ICES year–class index (YCI). The abundance (individuals/hour) of hake in these surveys is based on length.

For the purposes of analysis, only the individuals belonging to age group 0 were considered. This decision was based on the assumption that the individuals at that age have already passed the critical point of mortality. In this category, we included all hakes smaller than 20 cm in length (which ranged from 6 to 20 cm in length). In order to try to find a sign of stabilization of mortality, we subdivided this group into two subgroups. YCI was calculated as the sum of the abundance of hake according to two size categories:

YCI10 = sum of the abundance of individuals 6–10 cm in length.

YCI20 = sum of the abundance of individuals 11–20 cm in length.

The stock assessment also provides annual recruitment estimates (Recruit) for the entire stock per year (Table 3).

### 2.6. Temporal Variability in Ichthyoplankton Surveys

The temporal variability in the execution of surveys (see Table 1) may impact the EP and the proportion of small and large larvae reported in the area. The effect of the temporal variability of surveys is difficult to determine due to the year and time covariate. Figure 1 attempts to show the relationship between the residuals of SSB and EP according to the calendar day. The calendar day refers to the mean calendar day of the survey for each year. Figure 1 shows that the residuals of the linear regression are homogeneously distributed with the mean of the calendar day, which suggests that the differences seem to be more likely to be attributed to the annual variability of EP than to the survey date.

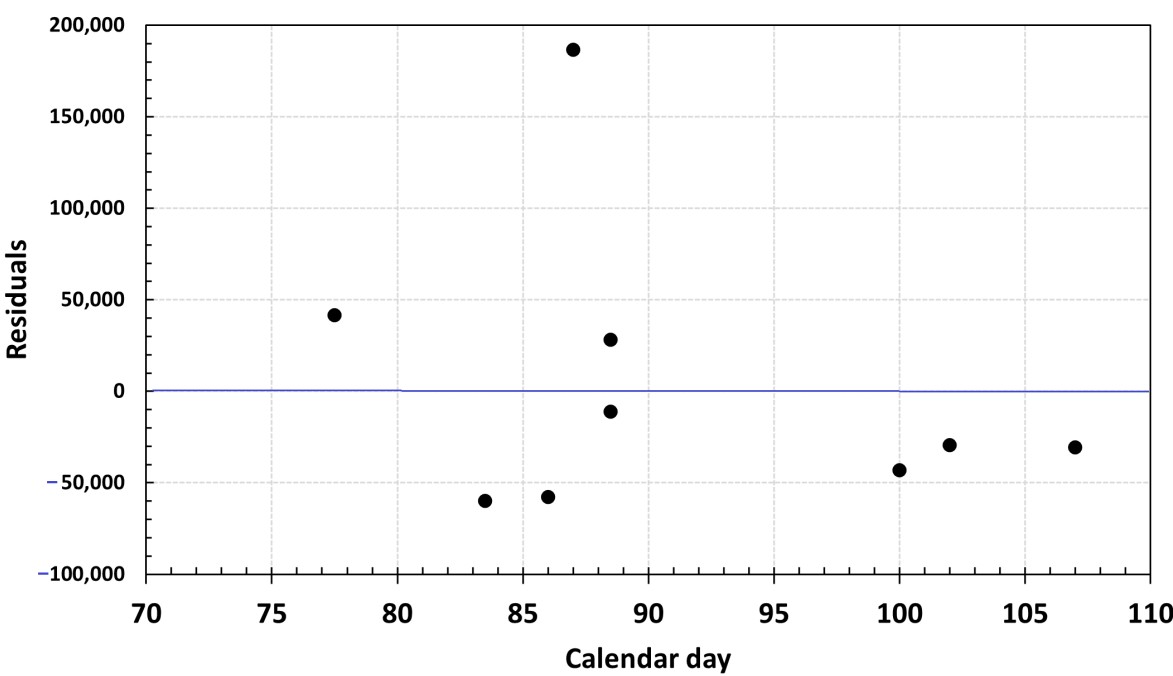

**Figure 1.** The plot of residuals of the linear regression between SSB and EP vs. the calendar day.

As for the larvae, we examined the larval size frequency to detect any influence of the survey date (Figure S3). The larval length varied from 2 to 13 mm SL, which are typical values for the type of nets being used in this study [40]. Moreover, we found that the contribution of larvae larger than 6 mm (N > 25) varied from 5% to 29% and was absent in 1998, in contrast to what was expected, as the survey started later (Table 1). We also estimated the mean SL of larvae for each year; we found that it varied from 3.8 mm

($\pm$0.65 mm) in 2016 to 5.3 mm ($\pm$0.87 mm) in 1995 independently of the survey date. In May, the presence of eggs and larvae in the area was neglected as, at that time, the spawning hake had shifted northward [27,28].

One of the points that can lead to controversy is the catchability of the different nets. In relation to this, and in the absence of more specific information, we have assumed that there are no differences between them. This is based on the fact that these nets are designed to sample a similar fraction of plankton.

*2.7. Statistical Analysis*

The information compiled from the different sources, i.e., ichthyoplankton surveys, the EVHOE database, and the stock assessment recruitment, was analyzed using the statistical software program Statgraphics (Statgraphics Centurion, Version XV, StatPoint, Inc.). This was performed according to the two following approaches: Linear regression with fitting each to the estimated abundance index (egg (EP) and larvae (N < 10 d, N > 10 d . . . . (see Table 2)) from the MEGS and Juveniles survey from the EVHOE campaigns (see Table 3)); and the Pearson correlation coefficient, which was estimated to measure the statistical relationship, or association, between the indices.

Finally, the influence points in the correlation were evaluated using the jackknife method [41]. The jackknife correlation measures the correlation between two variables removing the influence of single outliers, such as when $X = \{X_1, \ldots, X_N\}$ and $Y = \{Y_1, \ldots, Y_N\}$ are two random variables. The jackknife correlation consists of calculating the correlation between the two variables, removing one pair of observations at a time and taking the minimum. Mathematically, this can be expressed as

$$JC = min_{i=1}^{N}(O_i)$$

where $O_i$ is the correlation between variables $X$ and $Y$ without observations $X_i$ and $Y_i$.

**3. Results**

*3.1. Eggs and SSB Correlation*

Table 4 shows the estimates of hake egg production (EP), the total area surveyed, the positive areas (those with hake eggs), the mean temperature at a 20 m depth, and the percentage of a positive area for coverage between 46$°$–15$'$ N and 47$°$–45$'$ N. From 1995 to 2007, the EP gradually decreased with values fluctuating between 70 billion eggs in 1995 and 4 billion in 2007. The EP markedly increased to 250 billion eggs in 2013 and continued to rise in 2016, thereby achieving the highest value in the entire historical time series, three times more than in 2013. A further decline in abundance was observed in 2019.

**Table 4.** The annual evolution of the daily hake egg production estimates from 1995 to 2019. EP: Daily egg production; positive area: Areas with hake egg; $T_{20m}$: Temperature at 20 m depth; and SSB: Hake spawning stock biomass estimated in WGBIE [37].

| Year | Month | Total Area | Positive Area | EP (Egg/Day) | $T_{20m}$ | Positive Area | SSB |
|---|---|---|---|---|---|---|---|
| | | km$^2$ | km$^2$ | $\times 10^6$ | ($°$C) | % | ton |
| 1995 | 3 | 21,094 | 9261 | 25,972 | 11.9 | 44% | 57,940 |
| 1998 | 3–4 | 149,680 | 16,903 | 21,671 | 11.9 | 11% | 42,821 |
| 2001 | 3–4 | 90,549 | 10,490 | 14,931 | 12.0 | 11% | 51,936 |
| 2004 | 4 | 117,775 | 6320 | 6774 | 11.7 | 5% | 61,989 |
| 2007 | 4 | 71,523 | 2117 | 782 | 12.1 | 3% | 59,449 |
| 2010 | 3–4 | 77,982 | 10,549 | 9915 | 11.02 | 13% | 186,608 |
| 2013 | 3–4 | 103,434 | 23,117 | 155,830 | 11.28 | 22% | 259,818 |
| 2016 | 3–4 | 88,472 | 27,372 | 322,985 | 11.50 | 31% | 333,329 |
| 2019 | 3–4 | 103,079 | 25,176 | 41,969 | 11.63 | 24% | 298,571 |

The EP and positive area correlated positively (r = 0.737 and *p* < 0.05), although this relationship did not seem to be linear.

A high correlation (r = 0.76 and *p* < 0.04) was found between EP in subareas 8a–b and 8d and the estimates of northern stock hake biomass estimated by the WG assessment (Figure 2). However, it must be noted that this correlation increased to 0.95 (*p* < 0.05) when the last estimate (2019) was removed from the analysis.

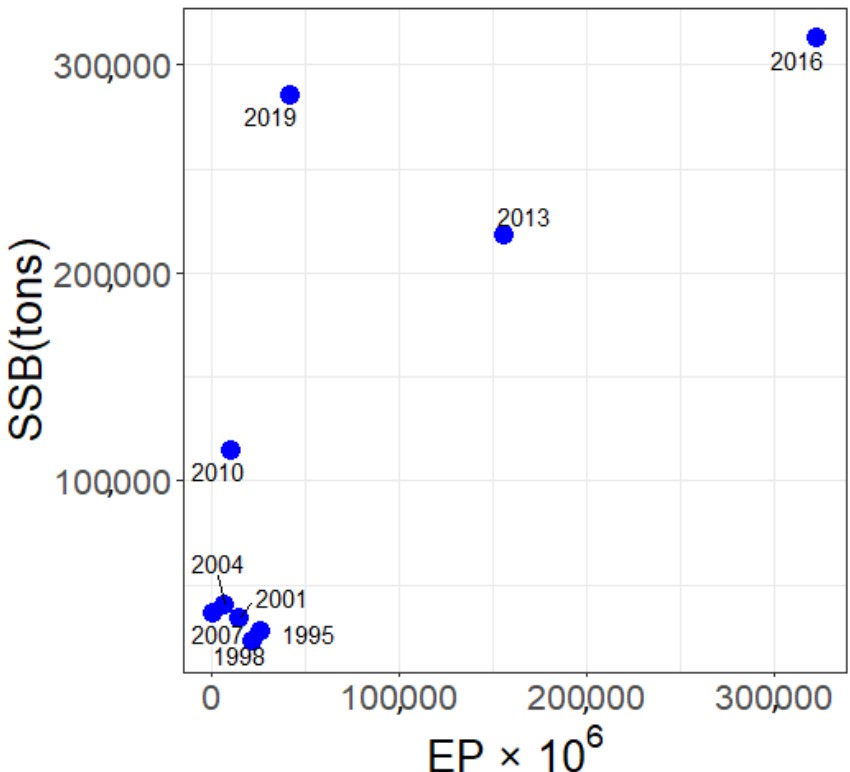

**Figure 2.** Graphic representation of hake eggs production (EP) estimated for the subarea 8abd and northern hake stock spawning biomass from WGBIE in subareas 4, 6, and 7, and in divisions 3a, 8a–b, and 8d [37].

In the influence point analysis, the observations that have a high influence on the correlation of two random variables are identified. The observation in 2016 has a more than three times greater influence on the correlation than the mean points. Furthermore, the jackknife correlation was equal to 0.64 and corresponds with the correlation obtained when the observation in 2016 is removed.

When the EP index obtained from the MEGS is plotted with the historical estimates of northern stock biomass (Figure 3), the good consistency of both time series is evident. It is also observed that for the years with low SSB (2001–2007), the egg index also presents low values, although the match to the SSB trend was less precise during this period.

*3.2. Larval Survival Estimates*

Exponential decay models were selected for fixing the data of mean larvae abundance (n°/m²) and length (SL, mm) by year. Further, Table 5 presents the estimated parameters for each model. All models were found to be statistically significant at the threshold of 95% or more, except for the years 1998 and 2007. Moreover, for years where the presence of larvae was low (1998, 2004, and 2007), the coefficient of determination was also low (Table 5).

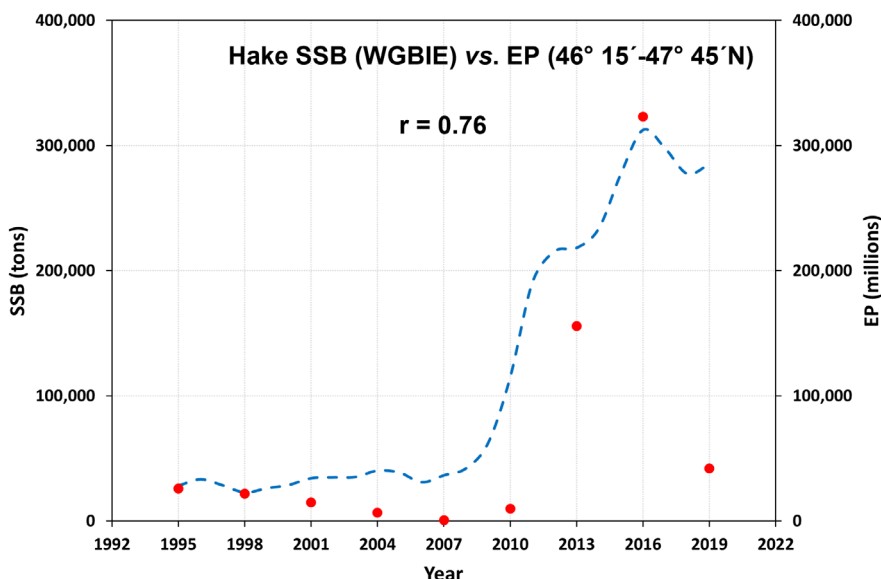

**Figure 3.** Historical estimations of the northern stock biomass of hake (SSB, tons) from the WG assessment (i.e., blue dashed line) in the subareas 4, 6, and 7, and in divisions 3a, 8a–b, and 8d (Source [37]) as well as egg production in the subareas 8a–b and 8d (i.e., red dots).

**Table 5.** The parameters of the mortality exponential model fitted to the data of hake larvae abundance (number/m$^2$) by 1 mm length (mm). SL: Average larval length; SD: Standard deviation; Lo: Larval abundance at hatch; Z: Instantaneous mortality; R$^2$: Coefficient of determination; $p$: Significant level; SvR: Survival rate; and T$_{20m}$: Temperature at 20 m depth (°C).

| Año | SL (SD) mm | $L_0/m^2$ | Z | R$^2$ (%) | $p$ | SvR | T$_{20m}$ |
|---|---|---|---|---|---|---|---|
| 1995 | 5.3 (0.87) | 150.08 | −0.651 | 88.9 | 0.0014 | 0.56 | 12.3 |
| 1998 | 4.0 (0.66) | 3.95 | −0.3365 | 26.2 | 0.0510 | 0.71 | 12.5 |
| 2001 | 5.0 (1.12) | 52.18 | −0.4957 | 52.1 | 0.0048 | 0.61 | 12.5 |
| 2004 | 4.1 (0.79) | 5.83 | −0.3245 | 34.4 | 0.0265 | 0.72 | 11.5 |
| 2007 | 5.0 (0.55) | 3.31 | −0.2983 | 24.4 | 0.0585 | 0.74 | 12.6 |
| 2010 | 4.3 (0.89) | 77.82 | −0.5765 | 80.8 | 0.0000 | 0.56 | 11.6 |
| 2013 | 4.8 (1.14) | 371.33 | −0.5451 | 73.1 | 0.0002 | 0.58 | 11.3 |
| 2016 | 3.8 (0.65) | 69.72 | −0.5812 | 66.3 | 0.0008 | 0.56 | 11.6 |

The mean survival rate is 63% per mm (CV = 12%), which turned out to be high from 1998 to 2007 (mean = 70%). However, this then decreased from 2010 to 2016 and was low in 1995 (mean = 56%). Furthermore, there was no statistical relationship found between the temperature and SvR. This is most likely due to the low-temperature variability between the surveys (CV of temperature = 4%).

*3.3. Egg, Larvae, and Recruit Correlations*

The correlation between EP and larva abundance at different development stages is shown in Figure 4. EP and small larvae at N < 10 days (SL < 3 mm and yolk-sac larvae) are positively correlated (R = 0.84 and $p$ < 0.05). However, there is no correlation between the abundance of yolk-sac larvae and other larval stages (N > 10 days old). A positive correlation was also detected, however, between the remaining larva age groups (N > 15, N > 25); nevertheless, the values of these correlations gradually decreased.

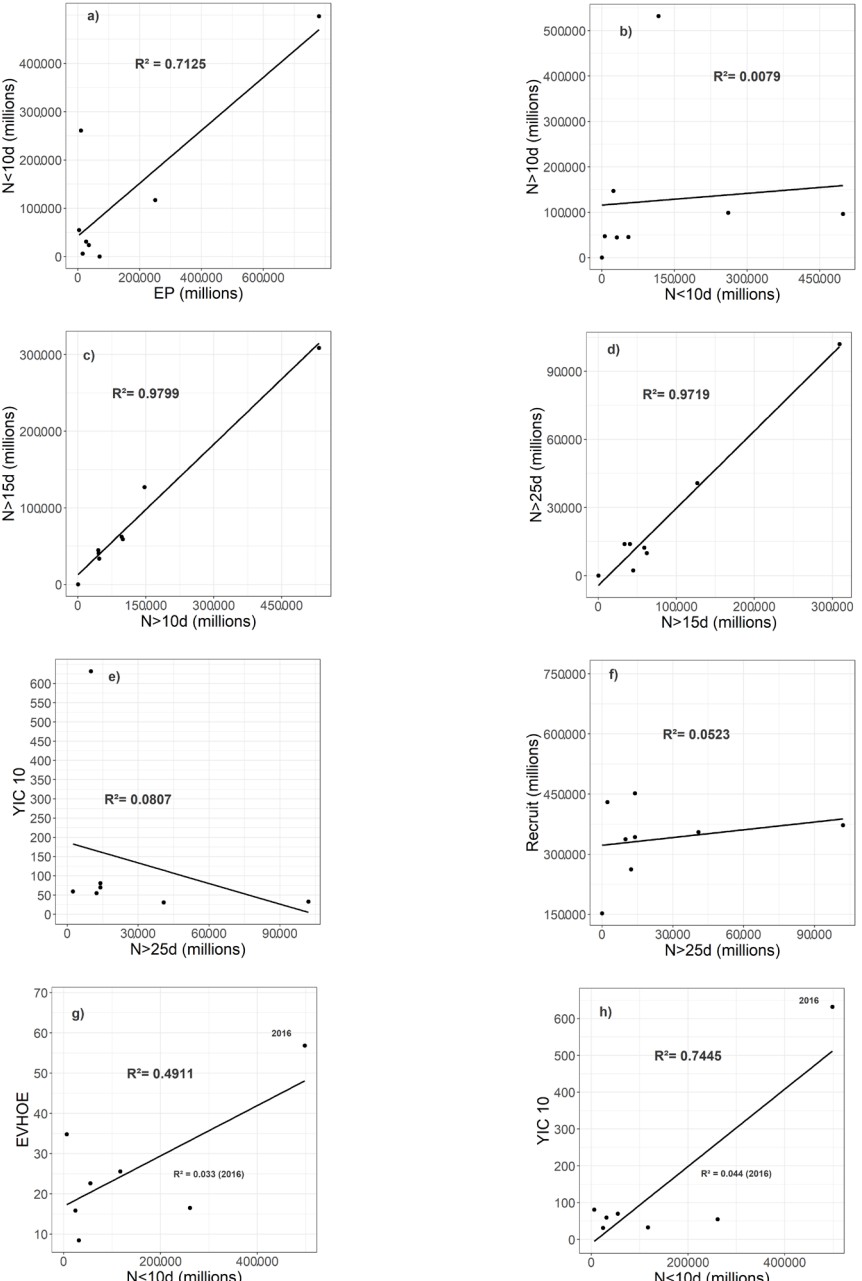

**Figure 4.** Correlation between (**a**) egg production (EP) and yolk-sac larvae (N < 10); (**b**) yolk-sac larvae (N < 10) and the other larvae stages (N > 10); (**c**) N> 15 vs. N > 10; (**d**) N > 25 vs. N > 15; (**d**) YCI 10 vs. N > 25; (**e**) YCI10; (**f**) recruits vs. N > 25; (**g**) EVHOE juveniles index vs. N < 10; and (**h**) YCI10 vs. N > 10. N >10: Abundance of larvae more than 10 days old; N > 15: Abundance of larvae more than 15 days old; N > 25: Abundance of larvae more than 25 days old; YCI10: Abundance of juveniles smaller than 10 cm (EVHOE surveys); and Recruit: SS3-derived age-0 recruitment [37]. In (**g**,**h**), the two $R^2$ values correspond to the estimate with all data ($R^2 = 0.49$ and 0.74) and without 2016 ($R^2 = 0.03$ and 0.04), respectively.

Table 6 shows the correlations between several larval abundance indices and recruit indices. The analysis included data from 1995 to 2016 for the recruit parameters and from 1997 to 2016 for the EVHOE data. This yielded a significant relationship of $R^2 = 0.49$ and 0.744 for the EVHOE index and the YC10 with N < 10, respectively. However, this correlation was strongly influenced by the year 2016 (Figure 4a), as the correlations were

not statistically significant when that year was removed ($R^2$ = 0.033 and 0.044, respectively) (Figure 4g,h).

**Table 6.** Correlation coefficients (bold and significant at $p < 0.05$) for the various indices of year–class strength and larval abundance (see Table 3 for acronyms).

|  | Recruit_BIE | EVHOE | YCI10 | YCI20 |
|---|---|---|---|---|
| $N < 10$ | 0.177 | **0.700** | **0.863** | 0.328 |
| $N > 10$ | 0.237 | 0.000 | −0.177 | 0.158 |
| $N > 15$ | 0.453 | −0.058 | −0.215 | 0.096 |
| $N > 25$ | 0.233 | −0.058 | −0.284 | 0.173 |

## 4. Discussion

### 4.1. Hake Egg Index as a Proxy for SSB Abundance

To the best of our knowledge, this is one of the few times when an egg index has been developed for a bentho-pelagic species [42], for Baltic cod). For pelagic species, such as Western horse mackerel stock, an egg index is currently used as an index for the relative abundance of SSB [43]. The use of an ichthyoplankton index minimizes the effort compared with the application of the daily egg production method (DEPM), due to the fact that no study of adult fertility is needed; furthermore, it provides a useful data source for the calibration of assessment models.

The applicability of a daily egg production method in assessing hake SSB was evaluated by Murua et al. [9] using abundance data obtained in the MEGS, and they concluded that the daily egg production method is potentially applicable to this stock. However, the correct application of DEPM to produce accurate estimates of the spawning population size requires an ad hoc research survey design to avoid systematic bias in the sampling of both ichthyoplankton and adults [9,44].

Although opportunistic studies can have serious limitations when applied to non-target species, they can demonstrate a good fit in some cases. The spatiotemporal distribution of hake spawning biomass in the Bay of Biscay was covered by the MEGS [11,26,28]. However, the coverage for which the data on hake eggs and larvae was available was heterogenous for each year in both cases, as well as regarding the place and time of their collection. These yearly differences can definitely have an impact on the final estimation of EP. The annual variability in the sampling coverage was therefore minimized by the process of standardizing in a common area; the differences in temporal execution of the surveys were shown to be negligible in an evaluation of the residuals of a linear model of the mean SSB and EP according to calendar day (Figure 1). Hake is a widely distributed stock and, as such, it is not easy to provide abundance indices that cover the whole stock distribution (which is what occurs with many other species). Indeed, the abundance indices currently included in the assessment only cover a limited part of the stock distribution but are considered a good unbiased estimator of recruitment or adult biomass.

We obtained a significant correlation between the SSB and EP in a standard area in the Bay of Biscay. This coefficient was determined, to a large extent, by the 2016 datapoint. Possessing a strong correlation between both variables is undoubtedly a good sign, as it suggests that both are good indicators of the stock abundance, particularly given the fact that it would otherwise be unlikely that both followed similar trends. However, a lower correlation does not invalidate the EP index; rather, it shows a divergence between the two sources that needs to be reconciled. The EP index is considered a good indicator of hake SSB and including it within the stock assessment model would provide additional information on mature individuals and allow one to obtain stock status estimates that are consistent with all data sources.

The early life stages of fish have been used to estimate adult abundance and the recruitment of different fish species. For species such as Atlantic mackerel (*Scomber scombrus*) and horse mackerel (*Trachurus trachurus*), ichthyoplankton samples have provided a measure of the relative abundance of adult biomass since the early 1980s [43,45]. In

regard to the Atlantic anchovy (*Engraulis encrasicoulus*) and sardines (*Sardina pilchardus*), the ichthyoplankton index has been applied for the purposes of stock assessment for more than 20 years [46,47]. In the Pacific, the CalCOFI ichthyoplankton time series have been used in stock assessments of rockfish and cowcod (*Sebastes levis*) [48] as well as of the Pacific or chub mackerel [49]. The use of these indices is essential for the annual assessment of the stock of these species. Further, we consider that it will also be of great value for hake.

Although the area considered only represents a small part of the total northern hake stock spawning area, the correlation with the SSB (for the whole area) shows that the index captures the SSB trends relatively well. The consistency between both time series can be explained by the fact that the Bay of Biscay is a relevant spawning ground for the stock. The abundance of hake eggs was determined with more extensive coverage in 1995 and 1998 [28]. Those results indicate a northward shift in the peak of hake spawning as the season progresses; on this note, two significant spawning grounds were also identified, one in the Bay of Biscay in March and the second in the Celtic Sea in April–May. The contribution of spawning grounds to the EP can vary annually, as is the case for other species, such as mackerel [50,51] or sardine [52], which may lead to a decreased correlation. However, at present, and with a time series from 1995, EP and SSB are still correlated, which reinforces the weight of the Bay of Biscay's spawning ground in the context of the northern stock of hake.

Regarding the abundance indices used to calibrate stock assessment models, it would be desirable to cover the entire range of distribution of the stock. In the assessment of the northern hake population, the three included contemporary abundance indices do not cover the entire area of stock distribution. For example, a slightly bigger area is covered by the recruitment index of the EVHOE survey than that captured by the egg/larvae indices (Figure S4). This is due to the fact that the former extends to the area toward the Celtic Sea, which is a well-known nursery area [28]. As the abundance indices are usually treated as a relative indicator of the abundance, the variability follows the same pattern in the entire distribution area, whereby not covering the whole area should not compromise the appropriateness of the index to be used in an assessment model.

Despite the limitations mentioned above, our results show that the MEGS could represent a valuable platform for the purposes of monitoring the status of the northern hake population through the egg production index. It would be desirable to extend the analyzed area and the participation of all institutes involved in the campaigns. In this regard, an important step has been taken with the recent inclusion of hake as a species of interest in the MEGS campaign manual.

### *4.2. Determination of the Year–Class Strength*

It is well known that a small adult population can exhibit high recruitment and vice-versa ([53] and Figure 5). The annual fluctuations in the recruitment of exploited fish populations have a major influence on stock size. Recruitment is mainly determined by the survival rates during the earliest life stages, namely egg, larvae, and juvenile [54]. Assuming that the mortality coefficient between the different life stages is constant, critical mortality points can be detected by analyzing how abundance changes between consecutive stages [21]. Hake eggs hatch in 6–8 days [33], and the newly hatched larvae start feeding after consuming their yolk in approximately 15 days [55,56]. The first feeding stage is documented as the critical point for larval survival and also, finally, for the purposes of recruitment, as the survival of a cohort at the end of the first year of life, which conditions the recruitment dynamics of fish populations [16]. Cushing extended the critical period, as defined by Hjort, to the entire larval development period, associating ocean physics as well as primary and secondary production to recruitment success. The positive correlation between the different stages of development from egg to larvae did not occur between yolk-sac larvae and the first-feeding larvae, or between the post-first feeding larvae and the YCI 10 (Figure 4). Our results indicate that the mortality between the yolk-sac and first-feeding larvae transitional period was sufficiently varied so as to ruin this relationship.

Similarly, the correlation between post-first feeding larvae and the recruit indices failed, as none of the tested recruitment indices could be used to establish the stage at which hake recruitment became fixed. This result strengthens the hypothesis that the year–class strength appears to be fixed later in the season. Watanabe et al. [21] did not identify any significant correlation between late larvae and the 1-year-old recruit for *Sardinops melanostictus* either. Furthermore, Houde [57] noted that small changes in mean vital rates in the later stages (50–70 days old) may have greater implications for recruitment than the loss of eggs and early larvae. By contrast, the early regulation of year–class strength has been proven for North Sea autumn-spawning herring [22,58] as well as for Baltic herring larvae at as early an age as 20–40 days old (20 mm length) [23].

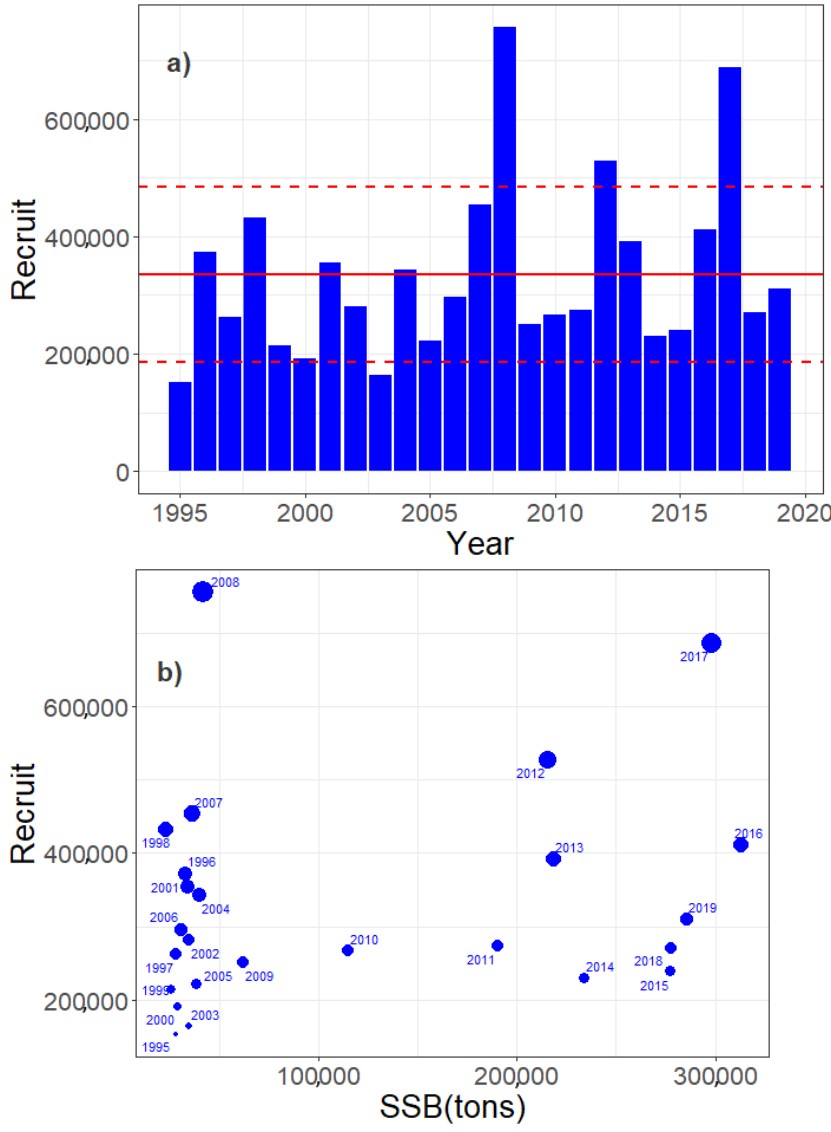

**Figure 5.** The (**a**) time series of recruitment and (**b**) the SSB–Recruit relationship of *M. merluccius* in the northern hake stock from 1995 to 2019. The red line indicates the mean recruitment value of the time series, and the red dashes represent the standard deviation. All the indicators are the output of the assessment model [37].

Our results suggest that year–class strength appears to be resolved some time after the two first months of life (i.e., the maximum age of the larvae in this study) and is most likely to be resolved in juvenile stages. However, the negative correlation between the larval survival rate and abundance could point to a certain density-dependent control during the

larval stages (Figure 6). Cannibalism in hake species is a well-known phenomenon that has been reported in both larval [59–61] and juvenile stages [62,63].

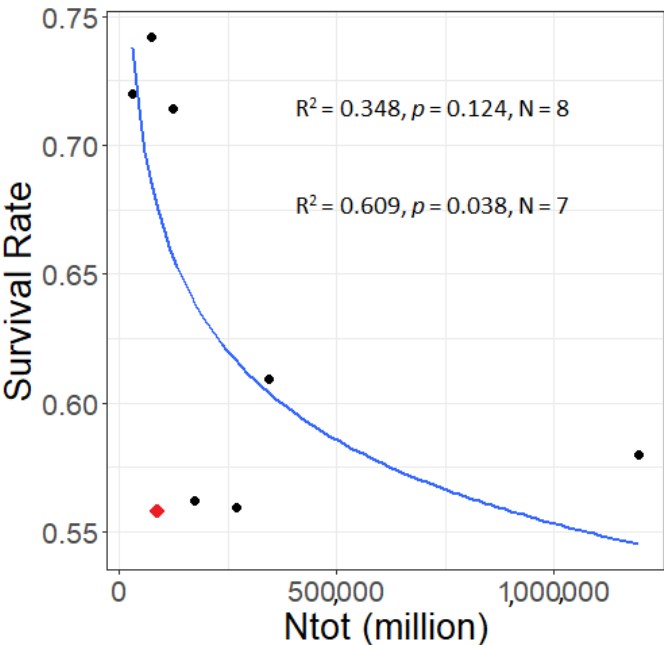

**Figure 6.** Relationship between larval survival (Survival) and total larval abundance (Ntot in millions). The power model best describes the relationship between these variables. The panel shows $R^2$ and the *p*. The relationship is statistically significant when the year 1998 (red point) is excluded.

## 5. Conclusions

With this study, we have demonstrated that long-term monitoring surveys of eggs and larvae can provide a reasonable index of relative SSB abundance as well as offer a simpler and cheaper alternative to the EP method assessed by Murua et al. (2010). We also shed light on the knowledge of early life stage dynamics in regard to the northern stock of hake. By connecting the abundance in the adjoining stages, we were able to identify two periods of high mortality associated with the transition of "yolk-sac-first feeding larvae" to "late larvae-YCI10", though we could not ascertain when year–class strength is determined. We propose that recruitment could be determined in the late juvenile stages and that cannibalism may be a mechanism that controls recruitment.

Despite the limitations discussed in the use of opportunistic surveys, our study has demonstrated the applicability of samples collected during ichthyoplanktonic campaigns for other purposes, thereby enabling value to be added to the samples by extending the analyses to other species. The significant correlation between the EP and SSB indicates, in our opinion, that the estimates of biomass and the egg index reflect what is happening in the population; therefore, continuing along this line would appear to be an effective option.

There are, nevertheless, still large gaps in the knowledge that should be improved. Both the assessment process and the ichthyoplankton campaigns certainly bear limitations, which may be often difficult to resolve. Monitoring plankton surveys should encompass the entire northern hake stock spawning season and area, which is currently being discussed by the WGMEGS [31]. This is also necessary for obtaining more realistic estimates and better identifying the conditions that control and regulate recruitment. As such, more effort should be made to investigate pre-recruit life stages, as recruitment can depend on the variability of survival during this phase.

**Supplementary Materials:** The following are available online at https://www.mdpi.com/article/10.3390/fishes8010050/s1, Figure S1: Times series of hake eggs in stage 1 (number/10 m$^2$) in the Bay of Biscay from 1995 to 2019. Dot sizes are proportional to egg abundance. Crosses indicate stations with no eggs. The square shows the standard area selected to estimate the production of hake eggs; Figure S2: Times series of hake larvae (number/10 m$^2$) in the Bay of Biscay from 1995 to 2016. Dot sizes are proportional to larvae abundance. Crosses indicate stations with no larvae; Figure S3: Spatial distribution of the abundance of juveniles of hake (<27 cm length, No indiv./hour) collected during EVHOE surveys for the time series (1996–2016), (Source ICES https://www.ices.dk/data/data-portals/Pages/DATRAS.aspx, accessed on 23 October 2022); Figure S4: Size frequency distribution (in %) of hake larvae collected from 1995 to 2016 grouped by 1 mm length.

**Author Contributions:** P.A.: Conceptualization, methodology, data curation, formal analysis, and writing of original draft; D.G.: Conceptualization, formal analysis, and writing; and U.C.: Conceptualization and supervision. All authors have read and agreed to the published version of the manuscript.

**Funding:** This research was partially funded by Spanish Ministry of Science and Innovation as part of the 2015 Framework Program on the Development and Innovation Oriented to the Challenges of Society through DREAMER—Recruitment Dynamics of European Hake Project (CTM2015-66676-C2-2-R) and by the Department of Agriculture and Fisheries of the Government of the Basque Country.

**Institutional Review Board Statement:** Ethical approval is not applicable for this article.

**Informed Consent Statement:** Not applicable.

**Data Availability Statement:** Not applicable.

**Acknowledgments:** The ichthyoplankton campaigns were conducted under the Framework of the EU Data Collection (DCF) through the ICES Group of mackerel and horse mackerel egg surveys. We thank all personnel involved in the collection and preparation of the data used in this work. This paper is contribution no. 1145 from AZTI.

**Conflicts of Interest:** The authors declare that they have no known competing financial interests or personal relationships that could have appeared to influence the work reported in this paper.

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
