# Peer review of "Investigating the Applicability of Ichthyoplanktonic Indices in Better Understanding the Dynamics of the Northern Stock of the Population of Atlantic Hake Merluccius merluccius (L.)"

_fishes, doi:10.3390/fishes8010050_

Round 1

Reviewer 1 Report

The manuscript evaluates the suitability of data from different surveys, e.g. mackerel and horse mackerel egg survey or EVHOE, for use in the assessment of the northern stock of Atlantic hake. Data on egg abundance are tested for use in SSB estimation, while data on larvae and juveniles were evaluated for use as a recruitment index.

The subject of the manuscript is interesting and well worth for publication if presented in an acceptable form.

However, in its current form, the manuscript is far from being acceptable for publication. I strongly recommend the authors to consult a native English speaker or at least someone capable of fluent English in writing. I consider this manuscript as a very early draft version not ready for submission to a journal for review. 

A few major comments to the text

lines 39-40: landing data cannot be considered as an indicator... This is not true. Landings data are and will be the primary source of data used in stock assessments! They have, admittedly, their limitations, which the authors also acknowledge. This is why independent survey data are used, mostly as tuning indices for assessment.

line 52: the authors mention 3 abundance indices in use for assessment without telling us precisely what data these are. Instead, they remain rather vague.

line 66, 67: herring is missing here. Early larvae abundance data are used in SSB estimation in the assessment for North Sea herring.

line 85: the first author's name is Oeberst!

line 86: a species which is both pelagic and demersal is called bentho-pelagic. Is there a reference for this statement? I suggest Heesen, H. and Murua, H. 2015. 31. Hakes (Merlucciidae) In: Heessen H, Daan N, Ellis JR (eds) Fish atlas of the Celtic Sea, North Sea, and Baltic Sea : based on international research-vessel surveys. Wageningen: Wageningen Academic Publ, pp 183-186

lines 117-119: there are some typos in the table 1. The German vessel is Walther Herwig III (abbreviation W. Herwig). The English one is CEFAS Endeavour (abbreviation Endeavour). The high speed samplers were not always Gulf IIIs. Germany used Nackthai samplers, England in 2004 a Gulf VII. Why is there a column with standard lengths given?

line 134: flowmeters were most probably not alway of GO brand. The Germans use Hydrobios. 

line 137: a reference to the current MEGS survey manual could be given here. It would be ICES 2019 (not the 2014 one as given in the references). At least the first author should know!

line 159: the formula for egg stage 1 duration has a typo: It should be 1264 (not 1275)

lines 191-195: this information should be put in table. Definition for the age classes should be stated a bit more precisely.

lines 197-201: why a length based mortality estimation when age conversions have been done. Only because it was a little more significant doesn't convince me. 

lines252-265: these 2 paragraphs belong in Material and Methods.

line 324: figure 3: Egg Production in tons? what does the figure show????

line 342,343: figure 4: Egg Production in tons, again? How is this calculated? 

line 374, 375: an egg production index is used for cod (a demersal species) in the Baltic. Köster, FW, Huwer, B, Kraus, G, Diekmann, R, et al. 2020: Egg production methods applied to Eastern Baltic cod provide indices of spawning stock dynamics, Fisheries Research, Volume 227, Article 105553

line 400 ff. the good correlation is mostly driven by just one data point - as the authors admit. I think the discussion on correlation and common trends could be a bit more substantial. What makes them believe that the egg index does provide a reliable index for SSB? Are more data needed in the time series?

line 446 ff. The authors discuss first feeding stage a the critical point in early life history of fish: It is one of a few more critical points in early life history. This is why eggs and early larvae abundance data do not enable us for recruitment predictions. 

line 464: the citation of Oeberst et al. 2009 is for Baltic herring larvae, not for North Sea autumn spawning herring, which Nash and Dickey-Collas 2005, and Payne et al. 2009 write about.

The discussion needs more structure and a thorough language check, as already stated above.

Author Response

Response to Reviewer 1 Comments

First of all, I would like to thank the reviewer for the timely suggestions received. We sincerely believe that they will greatly improve the work.

We hope that the corrections made will satisfy the reviewer's comments.

Reviewer 1

The manuscript evaluates the suitability of data from different surveys, e.g. mackerel and horse mackerel egg survey or EVHOE, for use in the assessment of the northern stock of Atlantic hake. Data on egg abundance are tested for use in SSB estimation, while data on larvae and juveniles were evaluated for use as a recruitment index.

The subject of the manuscript is interesting and well worth for publication if presented in an acceptable form.

However, in its current form, the manuscript is far from being acceptable for publication. I strongly recommend the authors to consult a native English speaker or at least someone capable of fluent English in writing. I consider this manuscript as a very early draft version not ready for submission to a journal for review. 

A few major comments to the text:

Point 1: lines 39-40: landing data cannot be considered as an indicator... This is not true. Landings data are and will be the primary source of data used in stock assessments! They have, admittedly, their limitations, which the authors also acknowledge. This is why independent survey data are used, mostly as tuning indices for assessment.

Response 1: We have modified a little bit the sentence. Line 37

Point 2: the authors mention 3 abundance indices in use for assessment without telling us precisely what data these are. Instead, they remain rather vague.

Response 2: A new paragraph has been included. Lines 60-66.

Point 2: line 66, 67: herring is missing here. Early larvae abundance data are used in SSB estimation in the assessment for North Sea herring.

Point 3: line 85: the first author's name is Oeberst!

Response 3:  It was a mistake that has been corrected.

Point 4: line 86: a species which is both pelagic and demersal is called bentho-pelagic. Is there a reference for this statement? I suggest Heesen, H. and Murua, H. 2015. 31. Hakes (Merlucciidae) In: Heessen H, Daan N, Ellis JR (eds) Fish atlas of the Celtic Sea, North Sea, and Baltic Sea : based on international research-vessel surveys. Wageningen: Wageningen Academic Publ, pp 183-186

Response 4: Thank you for the correction. We have modified the term, yet I could not consult the reference you suggested due to a lack of accessibility. As an alternative,  I have found in Murua, 2005 (PhD) the following paragraph: “The European hake is a demersal and benthopelagic species…. (section 3.4.2, pag 19) “.If the referee agrees I will use this as a reference. Line 101-106.

The term "benthopelagic" seems to refer to the feeding behavior of fishes, which feed on benthos and zooplankton. Most demersal fishes are benthopelagic.

Point 5: lines 117-119: there are some typos in the table 1. The German vessel is Walther Herwig III (abbreviation W. Herwig). The English one is CEFAS Endeavour (abbreviation Endeavour). The high speed samplers were not always Gulf IIIs. Germany used Nackthai samplers, England in 2004 a Gulf VII. Why is there a column with standard lengths given?

Response 5:Thank you for the correction. The table has been modified. Line 140

Why is there a column with standard lengths given?

Regarding the SL of the larvae in the last column, the reason is as follows: As the dates of the surveys are variable over time, the mean larval sizes could be influenced by this fact. So we decided to include this information in the table to make it easier for readers to visualize the variation of sizes from one year to another.

Point 6: line 134: flowmeters were most probably not always of GO brand. The Germans use Hydrobios

AND

Point 7: line 137: a reference to the current MEGS survey manual could be given here. It would be ICES 2019 (not the 2014 one as given in the references). At least the first author should know!. 

Response 6 and 7: In order not to be mistaken about the different brands of flowmeters, we have decided not to refer to any brand and only indicate Flowmeters. I hope this is enough for the reviewer.  We also have included the updated reference. Lines 180-183

Point 7: line 159: the formula for egg stage 1 duration has a typo: It should be 1264 (not 1275)

Respond 7: Sorry for the mistake, it is now corrected. Line 215

Point 8: lines 191-195: this information should be put in table. Definition for the age classes should be stated a bit more precisely.

Respond 8: We have slightly rewritten the text, included a formula and constructed a table following the reviewer's suggestion. Lines 240-255.

Point 9: lines 197-201: why a length based mortality estimation when age conversions have been done. Only because it was a little more significant doesn't convince me. 

Response 9: Another reason for choosing length over age in larval mortality models is that length is a less error-prone measure than age. The assignment of age is much more subjective than the determination of length, so we believe this contributes to the best of the models.  I have modified a little bit the paragraph with this explication. Lines 263-266.

Point 10: lines252-265: these 2 paragraphs belong in Material and Methods.

Response 10: Ok I moved the paragraphs to section 2.1 lines 184-192.

Point 11: line 324: figure 3: Egg Production in tons? what does the figure show????

Response 11: It was a mistake, we removed the unit. A new figure has been drawn. Thank you for the correction.

Point 12: line 342,343: figure 4: Egg Production in tons, again? How is this calculated? 

Response 12: It was a mistake, we removed the unit. A new figure has been drawn. Thank you for the correction.

Point 13: line 374, 375: an egg production index is used for cod (a demersal species) in the Baltic. Köster, FW, Huwer, B, Kraus, G, Diekmann, R, et al. 2020: Egg production methods applied to Eastern Baltic cod provide indices of spawning stock dynamics, Fisheries Research, Volume 227, Article 105553

Response 13: The reference is now in the text. Lines 514-517.

Point 14: line 400 ff. the good correlation is mostly driven by just one data point - as the authors admit. I think the discussion on correlation and common trends could be a bit more substantial. What makes them believe that the egg index does provide a reliable index for SSB? Are more data needed in the time series?

Response 14: We do not know if we have understood very well what the reviewer means by more substantial. I have added a paragraph referring to the assumption of "constant fertility" over time (Lines 559-569) and I think we have tried to demonstrate that despite the limitations derived from the data (time, area) the trend, EP and SSB, have remained over time. This gives a degree of robustness to the index. Ideally, the index should cover the entire hake spawning area, and retest its usefulness as a proxy of SSB tendency. This issue has already been addressed in the WGMEGS and hake has been listed as a species of interest. I have also added another paragraph at the end of the section (lines 639-644) referring to the latter.

Point 15:line 446 ff. The authors discuss first feeding stage a the critical point in early life history of fish: It is one of a few more critical points in early life history. This is why eggs and early larvae abundance data do not enable us for recruitment predictions. 

Response 15: I totally agree. We try to see if at some later larval stage we detect any signs of recruitment setting up, but according to our data, it seems that for hake this occurs much later in time, probably at the late juvenile stage.

Point 16:line 464: the citation of Oeberst et al. 2009 is for Baltic herring larvae, not for North Sea autumn spawning herring, which Nash and Dickey-Collas 2005, and Payne et al. 2009 write about.

Response 16: Thanks for the information, it has been added (Line 682).

Point 17:The discussion needs more structure and a thorough language check, as already stated above.

Response 17: Following the recommendations of both referees, we have made quite a few changes to the document. We have (1) included a new section in the results (2.6 Data analysis, line 346), (2) defined the acronyms and (3) reduced the number of graphs in figure 7 (we have left just one) and (4) modified slightly some paragraphs to improve the understanding.

The most important change concerns the deletion, in the Discussion, of the last paragraph, where the period was divided into two sub-periods and the differences between the periods of high and low hake biomass were analyzed for the different parameters (Table 6 and Figure 7 in the original paper). After some thought, this seems to us to be a bit out of place compared to the rest of the study. This would probably require an "ad hoc" study, with more data and analysis, which is far from the objective of this study. We believe that the study itself has sufficient data and results and its interest is not diminished by the deletion of this part; the same conclusions are obtained, except for those derived from that part of the analysis. I hope it will help your understanding. I also apologize for the grammatical errors and how this makes the text difficult to understand. The document has already been proofread by a native English speaker and I hope this will greatly improve your understanding.

Reviewer 2 Report

This manuscript addresses an interesting issue with regard to the development of an appropriate stock assessment approach for an important fisheries species.  I have two overarching impressions of this manuscript.  First, I believe the manuscript has some important information to present regarding the use of early-life history data in assessment of commercial fish stocks.  Second, but more unfortunately, the manuscript requires substantial revision, both in the text and the presentation of data.  Throughout all sections of the manuscript, there are issues of clarity, ambiguity, use of tense, undefined abbreviations/study site areas etc.  There are also issues with, more broadly, with the development of argument.  Together these limitations constrain a readers understanding of the research issue, aims,  methodological approaches, and interpretation of results.  While I believe the research team has some important data to present, a more strongly developed text is required to maximise the potential I think this manuscript represents.

Author Response

Point 1: This manuscript addresses an interesting issue with regard to the development of an appropriate stock assessment approach for an important fisheries species.  I have two overarching impressions of this manuscript.  First, I believe the manuscript has some important information to present regarding the use of early-life history data in assessment of commercial fish stocks.  Second, but more unfortunately, the manuscript requires substantial revision, both in the text and the presentation of data.  Throughout all sections of the manuscript, there are issues of clarity, ambiguity, use of tense, undefined abbreviations/study site areas etc.  There are also issues with, more broadly, with the development of argument.  Together these limitations constrain a readers understanding of the research issue, aims,  methodological approaches, and interpretation of results.  While I believe the research team has some important data to present, a more strongly developed text is required to maximise the potential I think this manuscript represents.

Response 1: We sincerely regret that we were not able to communicate our findings clearly enough for the reviewer to give us more concrete recommendations.

In any case, we have tried to improve the manuscript following the indications of one of the reviewers, which partly coincided with those expressed by this reviewer.

The changes we have made are: (1) Included a new section in the results (2.6 Data analysis, line 346), (2) defined the acronyms and (3) reduced the number of graphs in figure 7 (we have left just one) and (4) modified slightly some paragraphs to improve the understanding.

The most important change concerns the deletion, in the Discussion, of the last paragraph, where the period was divided into two sub-periods and the differences between the periods of high and low hake biomass were analyzed for the different parameters (Table 6 and Figure 7 in the original paper). After some thought, this seems to us to be a bit out of place compared to the rest of the study. This would probably require an "ad hoc" study, with more data and analysis, which is far from the objective of this study. We believe that the study itself has sufficient data and results and its interest is not diminished by the deletion of this part; the same conclusions are obtained, except for those derived from that part of the analysis. I hope it will help your understanding. I also apologize for the grammatical errors and how this makes the text difficult to understand. The document has already been proofread by a native English speaker and I hope this will greatly improve your understanding.

Round 2

Reviewer 1 Report

When I wrote in my first review that I consider the manuscript as a first draft, not ready for submission, and suggested that the authors should consider to consult someone capable in fluent English writing, I was hoping that the authors go through their manuscript meticulously and find and correct the many flaws themselves. Instead, the authors just answered the points I raised and also corrected the errors I pointed out. This is not what I would call a "major revision". Maybe I should have made myself more clear, and I apologise for not having done so.

The manuscript still contains many errors and issues, where the authors could be a bit more elaborate:

starting in line 58 you state that the assessment model lacks abundance indices of the biggest individual and that an egg index would be helpful here. Why do you think this is true? What do you mean with biggest individuals? What about the smaller individuals of the spawning stock. They produce eggs as well.

Line 81: recruitment is determined (not "takes place")

Line 95: this paragraph could be condensed as it contains some repetitions

Lines 102-103: evaluate utility of ichthyoplankton for calibration of the stock assessment model... this is not what you have done. You compared the index with assessment output w.r.t. similar trends. An evaluation for calibration of the model would include assessment model runs including an egg index and evaluate the output

Table 1: length data don't belong here. Why not include them in table 4?

Line 122: I would call these samplers just Gulf type. Gulf 3 is already a special versions (encased vs. unencased in Gulf 7 and Nackthai)

Line 135: this is a rather confusing paragraph. Consider rewording and condensing. Also in line 143 the area is defined in standard degrees and minutes while in line 176 it is given in decimal degrees (and rather confusingly noted in standard degrees and minutes)

Line 146: this paragraph needs rewording as well. 

Line 189/190 Larvae were aggregated into day-classes

Table 2 the definition of the age classes: N would refer to numbers not to age. Please use another letter. The definition of the length classes should be checked. E.g. Larvae ≤ 3 and the next one larvae ≥ 3.1 excludes all larvae = 3.0 mm. Better to give maximum and minimum lengths and ages for each class

Line 214: what do you mean with empirically expressed?

Line 216: for your model: did you use all nearest 0.1 mm below SL classes or just the 1 mm aggregated ones?

Line 222: what is SS3 model?

Line 228: did you use EVHOE data only? A reference for the DATRAS data is missing here.

Line 263: Hake show a protracted.... and both the beginning and end of the surveys .... have completely matched the main hake spawning period. - How can you say this? Is there any evidence?

Line 274 ff. what would be the catchability of the different larvae size classes w.r.t. the different samplers used and w.r.t. plankton samplers at all. 

Figure 2: the caption says that larvae were grouped into 1 mm length classes. Then, according to the labeling of the x-axis, there must have been larvae of -1 and 0 mm length. Did you mean 0.5 mm length classes?? N - the number of measured larvae behind these figures per year should be given.

Line 289 ff: in the text you say that all mortality models were significant while two p values in table 4 are > 0.05 ???

Figure 3 labelling of the x-axis: in the text you talk about billions of eggs white the labels show numbers in the 100,000s (e+05. see also table 5 where E+09 is given)

Figure 4: again, EP in e+05

Figure 5: EP is given in e+11

PLEASE CHECK THE SCALING OF YOUR FIGURES AND TABLES !!!!

Line 379: here you contradict just what you said in the sentence before: ... total fecundity shows high stability over time, while the opposite is observed for cod and hake (so, fecundity is not stable over time in hake). But then you conclude that impact of fecundity variability is expected to be negligible for the egg index (in hake!!)

Line 406 ff. you say that the assessment model could be wrong. Well, so could be the egg index!!

Figure 6: I cannot see the red line, nor the red dashes mentioned in the caption.

Line 447: are there some references for this statement?

Line 448: Annual fluctuations have a large influence on exploited fish populations. In an unexploited one, these fluctuations are not that important.

Line 454 ff: the first feeding stage is just one critical point of a few more. With that respect, your length range of captured larvae was certainly not enough to test for the use of larval abundances as recruitment indices. The cited indices for herring refer to larvae > 18 mm in the North Sea and > 20 mm in the Baltic. Age of the larvae in the North Sea is about 3-5 months.

Line 475: For larvae, cannibalism is not "well known" as it is based on just 1 observation (at least in that given reference). 

Line 487: references on cannibalism are missing here. 

Line 492: certainly?

Figure 7: How was this relationship established? Is that total larval abundance against mean survival over all length classes? Are the years included where the mortality model wasn't significant with a low r2? I'm afraid that this relationship is a bit far fetched.

Reviewer 2 Report

Dear Paula, Dorleta Garcia and Unai,

Thank you for providing a revised version of the manuscript for consideration.  This manuscript is much clearer (thank you) and I have read it with interest.  I think the manuscript provides some valuable information, and explores new opportunities for the assessment of commercial fish stocks.  you will see in my comments below, detail and comprehensive suggestions which I hope will help.

ABSTRACT:  The abstract covers the required information, but I have the following suggestions to increase clarity and conciseness.

L12-13: suggest wording change to: “Abundance indices have been typically derived from fishery dependent data, yet the recent increase in fisheries independent surveys now offers new opportunities for these calculations.”

L13-16: suggested wording change: “In this study we explored the usefulness of ichthyoplankton indices derived from scientific surveys to estimate spawning biomass, and if the year class strength of the commercial cohort of Atlantic hake could be defined at an early life stage.”

L16: delete “To this end” and “have” so it reads: “We used samples…..”

L18: the statement “ The suitability of MEGS surveys…” seems misleading in the context of the previous sentences that focus on the whether ichthyoplankton indices can be used to determine features of the commercial stock.  Suggest you delete this and start the sentence at “The biomass indices were determined…”

L19: the statement “…indices were determined as Stage 1 eggs….” Is unclear to me.  Can this be reworded for clarity?  How was this actually done?

L26: change to “…associated with the transition from…”

INTRODUCTION:

L32: change “approaches” to “datasets” and add full sop at end of sentence.

L33-38: these lines need to be reworded for clarity and flow.  Delete “based on the information obtained from fisheries” (L34); add reference to the statement of effectiveness of fishery dependent data (L35); integrate two sentences at L35; clarify the statement “dynamics of the fleets” (L36) clarify “has serious limitations” (L38, do you mean biases?).

L38-39: replace “Therefore” with “In contrast”. This sentence is critical in your introduction, it requires more development to clearly articulate what the importance/benefits are in using fishery independent data sources.

L40-42: This sentence appears out of place,  its relevance to the rest of the paragraph is unclear.  Remove, relocate or integrate.

L43-44: the difference between “catch and biological data” needs clarification.  A reference is required for this sentence.

L43-46:  the connection between these two sentences is unclear and requires clarification.  One sentence refers to data type whereas the next sentence refers to time series?  Delete the inclusion of “exploitation levels” and stay focused on the key aim to investigate stock abundance and year class strength

L43-48: reword, unclear and nonsensical.

L48: seems to be a new idea so insert paragraph break here.

L48-55: the start of this sentence refers to indices used in the stock assessment of hake, but the information that follows is essentially about surveys and associated datasets, rather than the indices.  Clarify.

L48-61: essentially these lines establish that knowledge of the abundance of commercial stock (big adults) is comprised by the available data; two of the three available datasets relate to recruitment stage size cohorts only.  Reword these lines to summarise this key fact.  At the moment it is hidden in detail surrounding the three datasets.

L70: delete “for stock assessment”

L62-74:  I would switch these two paragraphs. L68-74 should come before L62-67.  This change will better link with the final sentence of the preceding paragraph.

L76-77: unclear and vague.  Reword for clarity

L75-86:  the key message here can be strengthened.  As I read it the message is that early life stages exhibit elevated mortality after which mortality declines.  At this point, abundance of young animals is correlated to adult stock due to the reduced mortality in subsequent life.  Therefore knowing the earliest point at which this reduced mortality occurs could be useful as an index of commercial stock.  Rearticulate this paragraph more simply I suggest, this message is currently unclear.

Lines 89-90: delete “assessment”

L89-94: I don’t understand the relevance of this text.  Either clarify the relevance (i.e. need for greater certainty in stock assessments) or relocate it to the methods as a brief introduction to the fishery/fish.

Lines 92-94: clarify “expansion towards northern waters” (expansion of what?) and “higher increase” (increase of what?)

MATERIALS AND METHODS

L108: define ICES (first mention)

L110: change “and from” to “between”

L120-128: A map would be beneficial to international readers to display  some of the sampling area’s/grid you describe here. Suggest wording change to  “Most samples were collected by either a modified Gulf III type high speed sampler (15) fitted with a 250 μm mesh net or a 40 or 60 cm Bongo net with a 335 μm or 250 μm mesh size. Samples collected in 1995 were collected at the centre of 10 nm×30nm grid along transects perpendicular to the 200 m depth contour. At each station, a 126 plankton haul was made using a 60 cm Bongo net furnished with nets of 333 and 505μm 127 mesh size.”

L132: change “in every tow” to “during plankton tows”

L133: you mention material collected in the 250 μm net was processed and preserved in a 4% buffered formaldehyde solution. Clarify how samples collected using other methods were processed and preserved.

L150-151: The relevance of assigning eggs to four developmental stages in unclear.  Clarify or delete if it is unimportant.

L152: Change “eggs” to “egg”

L160-161: can this text be deleted and just refer to Equation 2 in the paragraph above? The notation for eq. 1 and eq. 2 seems odd.  E.g. why not have Eggs/m2/d and so on?

L172: clarify “positive area”

L174-177: this text repeats information at L135-143. Delete.

L222: Change “is conducted” to “was conducted”

L229: Change “indices” to “index”?

L231: clarify why only the individuals belonging to age group 0 were considered for analysis. Is this because this is the earliest stage at which you mortality has decreased to base-levels? And therefore that beyond this size/age, abundance give a fair representation of adult commercial stock? Similarly could you also clarify why you select YCI10 and YCI20.

L240-257: This section needs some more development in the description of analyses.  Can you be more specific in the describing what approach you applied to each of your objectives.  This would help the reader.

RESULTS

In general I think the results require some restructuring to provide clearer links to the two key objectives articulated at the end of the introduction.  I suggested focusing on key results that address the egg density index and larval cohort strength.  I would also suggest maintaining consistency in sequence of results.  Your aims first address egg density then larvae, so have this same sequence in the results.  This will help the reader as it is quiet a detailed paper so these ‘sign-posts’; will help I think. The relationship between section 3.2 and the aim appears unclear to me (I admit I am not a stock assessment specialist so maybe I have missed the point).  It may require some explanatory text that identifies what this section contributes to the aim.

L259 Section 3.1:  I think this should be summarized and placed in the methods section.  Essentially you are reporting some preliminary analyses that justify your methodological approach in selecting temporally variable datasets.  I think it is important information, but it detracts from the key results having it placed at the beginning of the results.

L283 and elsewhere: simply state the result the refer to the figure. It highlights the important result and reduces the word count

L283-285 and Fig 2: this needs further description.  Are the LFDs similar in each year? Could apply a KS test to statistically compare.  A sentence to clarify the importance of this presentation would be useful.

L291: clarify “all models” do you mean models constructed for each survey year?

L302: section numbering should be 3.3.  Spell out SSB in title

L308: delete “of”

L307-311 and Table 5: are the Eps reported here standardised by m2?  The equation indicates EP is, but the table indicates per day.  Clarify

Fig 3 and table 5: change sequence to match cross reference in text.

L366: correct cross reference, replace “Table 2” with “Table 3”

DISCUSSION

L375: define DEPM (first usage)

L402-409: I think it is good the influence of one year (2016) on the correlation between SSB and EP is discussed.  The influence of this year and its removal is large.  I think the this discussion requires more depth to adequately convince the reader the correlation offers something to the estimation of SSB.  Some references, or comparisons with of similar studies may help.  Some mention of the “valuable information not already included…” would be useful to also give greater understanding.

L413: BoB and “Bay of Biscal” inconsistently used.

Figure 6: I think this can be removed, it seems it simply demonstrates some key concepts you discuss.  Remove it I suggest

Figure 7: its not clear if this is your data?  If it is, then I should be present in the results, not the end of the discussion, or possibly added to the supplementary material.
